# Cone-Beam Computed Tomography-Derived Augmented Fluoroscopy Improves the Diagnostic Yield of Endobronchial Ultrasound-Guided Transbronchial Biopsy for Peripheral Pulmonary Lesions

**DOI:** 10.3390/diagnostics12010041

**Published:** 2021-12-25

**Authors:** Ching-Kai Lin, Hung-Jen Fan, Zong-Han Yao, Yen-Ting Lin, Yueh-Feng Wen, Shang-Gin Wu, Chao-Chi Ho

**Affiliations:** 1Department of Medicine, National Taiwan University Cancer Center, Taipei 106, Taiwan; vanhalen19781205@gmail.com (C.-K.L.); fnest.tw@gmail.com (H.-J.F.); med92leoyau@gmail.com (Z.-H.Y.); shwansiformosa@gmail.com (Y.-T.L.); b8501091@gmail.com (S.-G.W.); 2Department of Internal Medicine, National Taiwan University Hospital, Taipei 100, Taiwan; freeman0509@gmail.com; 3Department of Internal Medicine, National Taiwan University Hsin-Chu Hospital, Hsin-Chu 300, Taiwan; 4Department of Internal Medicine, National Taiwan University Biomedical Park Hospital, Hsin-Chu County 310, Taiwan

**Keywords:** cone-beam computed tomography-derived augmented fluoroscopy, diagnostic yield, endobronchial ultrasound-guided transbronchial biopsy, navigation success rate, peripheral pulmonary lesion

## Abstract

Background: Endobronchial ultrasound-guided transbronchial biopsy (EBUS-TBB) is used for the diagnosis of peripheral pulmonary lesions (PPLs), but the diagnostic yield is not adequate. Cone-beam computed tomography-derived augmented fluoroscopy (CBCT-AF) can be utilized to assess the location of PPLs and biopsy devices, and has the potential to improve the diagnostic accuracy of bronchoscopic techniques. The purpose of this study was to verify the contribution of CBCT-AF to EBUS-TBB. Methods: Patients who underwent EBUS-TBB for diagnosis of PPLs were enrolled. The navigation success rate and diagnostic yield were used to evaluate the effectiveness of CBCT-AF in EBUS-TBB. Results: In this study, 236 patients who underwent EBUS-TBB for PPL diagnosis were enrolled. One hundred fifteen patients were in CBCT-AF group and 121 were in non-AF group. The navigation success rate was significantly higher in the CBCT-AF group (96.5% vs. 86.8%, *p* = 0.006). The diagnostic yield was even better in the CBCT-AF group when the target lesion was small in size (68.8% vs. 0%, *p* = 0.026 for lesions ≤10 mm and 77.5% vs. 46.4%, *p* = 0.016 for lesions 10–20 mm, respectively). The diagnostic yield of the two study groups became similar when the procedures with a failure of navigation were excluded. The procedure-related complication rate was similar between the two study groups. Conclusion: CBCT-AF is safe, and effectively enhances the navigation success rate, thereby increasing the diagnostic yield of EBUS-TBB for PPLs.

## 1. Introduction

With the increasing use of low-dose computed tomography (CT) for lung cancer screening [1], peripheral pulmonary lesions (PPLs) are more easily exposed. For suspected malignant PPLs, an accurate diagnosis is an essential step in devising an appropriate treatment plan. The transthoracic approach with CT-guided biopsy traditionally is the first choice due to its having the highest diagnostic accuracy [2]. However, because of its high complication rate, which may lead to patient morbidity and mortality, the use of bronchoscopic techniques has gradually increased.

Endobronchial ultrasound-guided transbronchial biopsy (EBUS-TBB) is now widely used for the diagnosis of PPLs [3,4,5]. In previous publications, the diagnostic accuracy of EBUS-TBB alone has ranged from 60–70% [6,7,8]. Combined use with other methods, such as fluoroscopy or virtual bronchoscopic navigation, has been attempted to improve diagnostic accuracy [9,10,11]. However, these techniques have their limitations or disadvantages, so a more effective system is required to assist EBUS-TBB procedures.

Cone-beam CT (CBCT) is a newer CT modality that can provide both real-time 2-dimensional (2D) fluoroscopy and 3D CBCT scans. With dedicated software, the target can be contoured, and be projected onto live fluoroscopy images, termed augmented fluoroscopy (AF). This system provides real-time information for interventional radiologists and surgeons in many advanced procedures [12,13,14,15,16,17]. However, there is scant evidence as to the utility of combining EBUS-TBB and CBCT-derived AF (CBCT-AF) in the diagnosis of PPLs. Our aims in this study were to investigate the effectiveness of CBCT-AF in EBUS-TBB, and to evaluate the possible reason for CBCT-AF’s effect on the diagnostic yield.

## 2. Materials and Methods

### 2.1. Participants

This was a retrospective chart review of patients who underwent EBUS-TBB for PPL diagnosis at the Department of Thoracic Medicine, National Taiwan University Cancer Center and National Taiwan University Hsin-Chu Hospital, from January 2018 to June 2020. Each patient underwent an EBUS-TBB procedure once, and only one target lesion was sampled in each procedure. 

Patient data regarding age, gender, and final diagnosis were collected. In order to characterize the EBUS-TBB procedure, the following data were also recorded: the indication (initial diagnosis vs. re-biopsy), lesion size (length of the long axis measured on the CT image), lesion pattern (solid vs. part-solid/ground glass opacity (GGO) on the CT image), presence or absence of a bronchus sign, lesion location, visibility of the lesions on chest plain film, guide sheath use, location of the probe (within vs. adjacent to/invisible), procedure time (interval between insertion and removal of the bronchoscope through the vocal cords), procedure-related major adverse events, and radiation dose. 

The study population was divided into two groups depending on the use of CBCT-AF. We defined the “CBCT-AF group” as those that underwent CBCT-AF combined with EBUS-TBB. The “non-AF group” included those for whom CBCT-AF was not performed during the EBUS-TBB procedure. Written informed consent was obtained from each patient prior to bronchoscopy. The study was approved by the National Taiwan University Cancer Center Institutional Review Board in 8 April 2021 (IRB # 202102061RINC).

### 2.2. Procedures

All bronchoscopy procedures were performed by our pulmonologist (C.K.L.), who had more than 10 years of experience in EBUS-TBB. All EBUS-TBB procedures in the non-AF group were performed at National Taiwan University Hsin-Chu Hospital, depending on the general bronchoscopy room setting. All CBCT-AF procedures, with or without intra-procedural CBCT procedures (CBCT-AF group), were performed in the hybrid bronchoscopy room with a C-arm CBCT angiography system (Artis Zee Ceiling; Siemens Healthcare GmbH, Forchheim, Germany) at National Taiwan University Cancer Center.

Each patient in the CBCT-AF group was placed on the angiographic table in the supine position before EBUS-TBB. To acquire the 3D dataset, the C-arm needed to rotate 200° (180° plus the fan angle) around the target, maintaining an inspiratory hold. A radiological technician highlighted the area of the target lesion and target bronchus using annotation software (syngo iGuide Toolbox; Siemens Healthcare GmbH, Forchheim, Germany). The annotated marker was then projected onto the 2D-fluoroscopy live screen in accordance with the corresponding 3D locations; thus, the AF system was configured.

After premedication with lidocaine local anesthesia and intravenous midazolam/propofol and fentanyl for conscious sedation, all patients in both groups underwent the EBUS-TBB procedure using a flexible bronchoscope (BF-Q290 or BF-P290; Olympus Co., Tokyo, Japan) combined with a 20 MHz radial-EBUS (UM-S20-17S; Olympus Co., Tokyo, Japan). In the non-AF group, the radial-EBUS was inserted through the working channel of the scope into the suspected target bronchus, based on CT findings and without a navigation system. In the CBCT-AF group, radial-EBUS was guided by AF imaging. After confirming the location of the lesion using radial-EBUS imaging (defined as “success of navigation”), with or without intraprocedural CBCT, TBB with forceps (NBF01-11018120; MICRO-TECH Co. Ltd., Nanjing, China) or a commercial GS kit (K201; Olympus Co., Tokyo, Japan) was carried out for specimen collection. Rapid on-site cytologic evaluation (ROSE) was always available for confirming lesion access. The material from the forceps biopsy was imprinted on a clear glass slide, and then stained using a rapid method (Hemacolor; Merck KGaA, Darmstadt, Germany) for ROSE. If neither a malignant cell nor a specific finding was noted by the ROSE study, we would change to another site for biopsy. The procedure would be terminated when there was no specific finding more than 3×, or if the patient could no longer tolerate the procedure. The histologic samples were placed in 10% formalin and were embedded in paraffin for routine histologic evaluation with hematoxylin and eosin staining, and were interpreted by our cytopathologists. Tissue cultures of bacteria, mycobacteria, and fungus would also be added when an infectious PPL was suspected. If radial-EBUS was unable to detect the target lesion, or radial-EBUS unable to arrive to the target lesion that confirmed by intraprocedural CBCT (meaning “navigation failure”), no further TBB would be performed. Neither brushing nor transbronchial needle aspiration was performed during the procedure. Figure 1 shows the flowchart of the procedure in CBCT-AF and non-AF groups, and Figure 2 shows imaging results for a lung cancer patient diagnosed by CBCT-AF and EBUS-TBB.

### 2.3. Diagnosis

The final diagnosis of PPLs was established based on cytopathologic evidence, microbiological analyses, or clinical follow-up. Benign inflammation, which could not be determined cytopathologically or microbiologically, were confirmed by radiological and clinical follow-up (unchanged or decreased lesion size on the CT image) at least 6 months after bronchoscopy. Malignant PPLs underwent re-biopsy after cancer treatment, and malignancy with disease progression was still considered if the lesions were enlarged, as seen in the radiological follow-up, and if an infection/inflammation process was excluded, even if there was no histological confirmation.

### 2.4. Statistical Analysis

We evaluated the effectiveness of CBCT-AF in EBUS-TBB based on the navigation success rate and diagnostic yield. The navigation success rate was defined as a “successful navigation class/total testing class”. The diagnostic yields were defined as “correctly TBB-proved class/total testing class”. For malignant PPLs, “correctly TBB-proved class” means that positive of cytologic or pathologic report revealed by TBB specimens. Suspicious findings were considered nondiagnostic in our study. For infectious PPLs, “correctly TBB-proved” needs positive tissue culture result via TBB specimen if no microorganism is detected from histologic report. The “correctly TBB-proved class” was also mean true positive. When “chronic inflammation” was revealed via histologic report, “true negative” would still be considered if no specific infection/inflammation process was found, and the PPLs decreased in size or disappeared within 6 months follow-up without any special treatment. False negatives were defined as TBB samples revealed negative malignant histologic report initially, and malignant cells discovered later through repeated biopsy via any biopsy methods, or disease progression confirmed via radiological and clinical follow-up. The sensitivity, specificity, positive predictive value (PPV), negative predictive value (NPV), and diagnostic accuracy were calculated via standard definitions.

Comparisons were made using Student’s t test or one-way analysis of variance (ANOVA) for continuous variables, and the ꭓ^2^ test or Fisher’s exact test for categorical variables. A *p*-value of less than 0.05 was considered significant. We used SPSS version 21.0 (IBM, SPSS, Chicago, IL, USA) for statistical analysis.

## 3. Results

### 3.1. Patients and Target Lesions

In all, 236 patients who underwent 236 EBUS-TBB procedures for PPL diagnosis were enrolled during our study period. One hundred fifteen patients (procedures) were in CBCT-AF group and 121 patients (procedures) were in non-AF group. The mean age of our total study population was 66.1 years, and 50% of the patients were male. Malignant diagnoses were common (73.3%) in our study population, and most of them were primary lung cancer. The CBCT-AF group had a higher proportion of malignancy. The details of the final diagnoses of both groups and the corresponding cytopathologic results are shown in Table 1 and Table 2.

In the present study, the lesion size was smaller in the CBCT-AF group (24.0 mm vs. 30.6 mm, *p* < 0.001). The proportion of lesions with a solid appearance was lower in the CBCT-AF group than in the non-AF group (76.5% vs. 86.8%, *p* = 0.030). Fewer patients in the CBCT-AF group used a guide sheath than in the non-AF group (39.1% vs. 65.3%, *p* < 0.001). Longer procedure duration was also noted in the CBCT-AF group (41.9 min vs. 34.4 min, *p* < 0.001). There were no statistically significant differences between the 2 groups in terms of the indication of the procedure, location of the target lesions, presence of a bronchus sign, location of the radial-EBUS probe, visibility of the lesions on chest plain film, and procedure-related complications. No serious respiratory or hemodynamic adverse event was documented (Table 3).

In the CBCT-AF group, intra-procedural CBCT was more frequently used for smaller target lesions (≤10 mm vs. 10–20 mm vs. >20 mm, 37.5% vs. 25% vs. 5.1%, respectively). The mean radiation dose for fluoroscopy was 2.70 Gy∙cm^2^, and the Dyna-CT dose was 22.78 Gy∙cm^2^. The mean total radiation dose was 25.48 Gy∙cm^2^.

### 3.2. Navigation Success Rate and Diagnostic Yield

The CBCT-AF group had a higher navigation success rate than the non-AF group (96.5% vs. 86.8%, *p* = 0.006). In the subgroup analysis, the effect of CBCT-AF on the navigation success rate was significant in patients with a small lesion size (CBCT-AF vs. non-AF: 87.5% vs. 0%, *p* = 0.003 in the subgroup with lesions ≤ 10 mm, and 95% vs. 71.4%, *p* = 0.009 in the subgroup with lesions 10–20 mm in size, respectively) (Table 4).

Smaller lesions also had a higher diagnostic yield in the CBCT-AF group. When the lesion size was ≤10 mm, the diagnostic yield was 68.8% (11/16) in the CBCT-AF group, and no patient (0/4) with a lesion that size was successfully diagnosed in the non-AF group (*p* = 0.026). The diagnostic yield of EBUS-TBB for lesions 10–20 mm in size in the non-AF group was only 46.4% (13/28). The diagnostic yield was significantly elevated to 77.5% (31/40) when CBCT-AF was performed with EBUS-TBB (*p* = 0.016) (Table 4). The performance of the diagnosis of PPLs between CBCT-AF and non-AF groups were in Table 5.

### 3.3. Population Excluding Navigation Failure

After excluding 20 procedures (4 in the CBCT-AF group and 16 in the non-AF group) that were navigation failures, the remaining 111 procedures in the CBCT-AF group and 105 procedures in the non-AF group underwent further analysis. The overall diagnostic yield was similar between the 2 groups (86.5% vs. 85.7%, *p* = 0.513). When the target lesion size was 10–20 mm, the diagnostic yield was still not significantly different between the 2 groups (81.6% vs. 65%, *p* = 0.141). The diagnostic yield for other characteristics (indication, lesion appearance, lesion location, bronchus sign, position of the probe, guide sheath use, and final diagnosis with malignancy) was equal between the 2 study populations (Table 6).

In addition to CBCT-AF use, we found that lesion size might also influence the diagnostic yield of EBUS-TBB in the subgroup analysis. In the CBCT-AF group, large lesions (>20 mm) had a higher EBUS-TBB diagnostic yield than small target lesions (10–20 mm) (91.5% vs. 81.6%, *p* = 0.048). This was also found to be true in the non-AF group (90.6% vs. 65%, *p* < 0.001). No other factor has been detected to affect the diagnostic yield of EBUS-TBB.

## 4. Discussion

This retrospective study found that the navigation success rate and diagnostic yield were improved when CBCT-AF was performed during EBUS-TBB of lesions ≤20 mm. After excluding the procedures that were a failure of navigation, the diagnostic yield showed no difference between the CBCT-AF group and the non-AF group.

TBB has never been a simple procedure for the diagnosis of PPLs. From the technical standpoint, at least three major steps influence the bronchoscopic technique. The first step is demonstrating the capability of reaching a target (navigation). The next step is demonstrating the arrival of the sampling device at the target (confirmation). The final step should be obtaining adequate diagnostic samples (acquisition) [18,19]. Radial-EBUS plays an important role in the “confirmation” step, by verifying contact with the target through its real-time imaging. ROSE provides immediate feedback to ensure proper samples for morphological analysis, thereby implementing the “acquisition” step [8,20]. For the “navigation” step, virtual bronchoscopic navigation or electromagnetic navigation have been considered to be traditional guiding tools in bronchoscopic techniques. Because of the use of static CT imaging, it is impossible to compensate for the motion effect of breathing [21]. Using the navigation system alone for the TBB procedure yields an unacceptable navigation success rate. The use of fluoroscopy has also been attempted to guide TBB [9]. Using real-time imaging, operators can immediately realize the bronchial route through correlation between the target lesion and the tool. Although small or GGO lesions are not easily visible under fluoroscopy [22,23], CBCT-AF could provide the integrated tracking that is derived from 3D CBCT data, which helps the operators easily identify these invisible lesions on the fluoroscopy image. In the present study, CBCT-AF significantly improved the navigation success rate in EBUS-TBB (CBCT-AF vs. non-AF, 96.5% vs. 86.8%, *p* = 0.006). The influence of CBCT-AF on the navigation success rate was significant in the small lesion population (lesion size ≤20 mm). CBCT-AF seems to play a navigation role in the EBUS-TBB procedure.

To our knowledge, it was very difficult to diagnose subcentimeter lesions (≤10 mm) via R-EBUS and navigation system. In the present study, the diagnostic yield in the CBCT-AF group was higher than that in the non-AF group when the lesion size was ≤10 mm (68.8% **vs.** 0%, *p* = 0.026). Not only malignancy but also benign disease can be successful diagnosis (Table 7). In the study population with a lesion size of 10–20 mm, the CBCT-AF group also had a better diagnostic yield (77.5% vs. 46.4%, *p* = 0.016). This finding has been reported previously [24]. Pritchett et al., reported that combined electromagnetic navigation and CBCT-AF had a high diagnostic yield for biopsy of PPLs [25]. Casal’s study of 20 consecutive patients found that utilization of intraprocedural CBCT helped improve the diagnostic accuracy of the thin/ultrathin bronchoscope, and radial-EBUS for PPLs [26]. Overall, the use of CBCT significantly improved the bronchoscopic technique in the diagnosis of PPLs.

There was no difference in the diagnostic yield between the two groups when we omitted the procedures with a failure of navigation. This indicates that the effect of CBCT-AF is eliminated when the factor of navigation is removed. We confirmed that the effect of CBCT-AF in improving the diagnostic yield of EBUS-TBB is mainly due to its assistance in navigating to the target lesion.

Several studies have tried to evaluate the possible navigation benefit of CBCT in performing the bronchoscopic technique [21,26,27]. Most of the studies combined CBCT with virtual bronchoscopic navigation or electromagnetic navigation in their procedures [19,28,29,30], so it would not be clear whether the benefits were due to CBCT or to the navigation system. Furthermore, few of these studies had a control group. In the present study, we used only CBCT-AF as the navigation tool. In addition to enrolling non-AF patients as our control group, we also included 115 patients in our CBCT-AF group. To our knowledge, this is the largest CBCT-AF study population up to now. We believe our results are convincing and will be helpful in clinical practice.

CBCT-AF has the ability to evaluate “tool-in-lesion” by using real-time 2D imaging. Intraprocedural CBCT offers even better 3D information for confirmation [25,28]. The diagnostic yield in our study was significantly reduced when the target lesion was less than 20 mm in size. Intraprocedural CBCT was attempted more frequently to increase the diagnostic yield in the small lesion population. However, this condition persisted in both groups, even though the study population with a failure of navigation was not included. The cause of this could be the motion effect on the lesion due to respiration [31]. In Chen’s study, the divergence due to the motion effect could be more than 20 mm. The projected target from AF and CBCT imaging can only provide static information. It is very difficult to ensure that the biopsy device is inside the small lesion during breathing activity. CBCT-AF remains unsatisfactory in terms of overcoming the diagnostic impact of small lesion sizes.

The overall diagnostic yield of EBUS-TBB was not significantly different between the CBCT-AF and non-AF groups in all study populations (83.5% vs. 74.4%, *p* = 0.060). We believe this was because of the components of the study population. There was a large proportion of lesions with a size >20 mm in the non-AF group (73.6%, 89/121). The power of navigation via CBCT-AF was then weakened in this situation.

Studies have found that some factors, such as re-biopsy of post-treated lesions, non-solid lesion appearance on CT, upper lobe location, absence of a bronchus sign, probe adjacent to the lesion, visibility of the lesions on chest plain film, guide sheath use, and final diagnosis of a malignant process might affect the diagnostic yield of EBUS-TBB [5,32,33,34,35,36,37]. In the present study, there was no significant difference in the diagnostic yield in the subgroup analysis (Table 5). These results are very similar to that of our previous publication [8]. By including ROSE in the procedure, the diagnostic yield of EBUS-TBB was not only improved, but the effect of the factors mentioned above was reduced. In this situation, the interference of non-homogeneous population in the present study could be decreased.

The duration of the procedure in the CBCT-AF group was longer than that of the non-AF group (41.9 vs. 34.4 min., *p* < 0.001). This is most likely due to the increased time needed for AF and CBCT during the procedure. Of note, the overall complication rate was not elevated in the CBCT-AF group, and the incidence of post-procedure-related fever and bleeding was not statistically higher in the CBCT-AF group than in non-AF group. The three patients that developed pneumothorax after TBB were in non-AF group. We believe the real-time imaging provided by CBCT-AF also helped us avoid injury to the pleura. There was also additional radiation exposure in our study population, which might cause some complications, such as skin injury or risk of cancer [38,39,40]. In the present study, the mean total radiation dose, Dyna-CT dose, and radiation dose for fluoroscopy were 25.48, 22.78, and 2.70 Gy∙cm^2^, respectively. The mean estimated dose was 4.1 mSv for total radiation dose, 3.6 mSv for Dyna-CT and 0.4 mSv for fluoroscopy, using a generalized conversion factor of 0.16 mSv/ Gy∙cm^2^. Very low risk of increasing lifetime fatal malignancy was considered when the exposed radiation dose was lower than 10 mSv [41]. No patient obtained radiation induced dermatitis or skin damage after the procedure in our study population. We believe CBCT-AF was relatively safe, in terms of both procedure-related complications and radiation exposure dose.

All histologic report with “chronic inflammation” has been considered nondiagnostic in some publications [7]. In the present study, positive tissue culture via TBB is an essential criterion for “correctly TBB-proved” in infectious PPLs if no microorganism or only chronic inflammation was revealed in the formal histologic result. When PPLs without specific infection/inflammation process was determined initially, repeat invasive diagnostic methods (repeat bronchoscopy, CT-guided biopsy or surgical resection) would be performed. Only the PPLs decreased in size or disappeared within 6 months follow-up and did not receive special treatment were considered “benign inflammation”, while TBB was considered “true negative”. We believed this definition might be closer to the clinical situation.

Our study has several limitations. First, this was a retrospective study. Some clinical information, such as navigation time, duration of biopsy, and the number of ROSE slides used in each procedure, was not mentioned in the chart review. These factors might also have represented the navigation success rate and diagnostic accuracy of EBUS-TBB in our study. Second, some characters are different between 2 of study population (lesion size, lesion appearance and guide sheath use). The difference could be minimized by using some methods such as propensity score matching. However, the positive findings that are actually meaningful could be missed. For example, only four lesions in non-AF group are smaller than 10 mm. Most small lesion population in CBCT-AF group might also be excluded if propensity score matching was used in our study. Therefore, a prospective randomized study is warranted. Third, fewer benign PPLs were enrolled in our study population (20% in CBCT-AF group and 33.1% in non-AF group). This may cause a lower population of true negative, thereby cause a lower NPV.

Fourth, there was only one operator performing the procedures might hamper the generalizability of the results, but this also eliminates the factors of the learning curve and experience of different operators. Otherwise, our overall diagnostic yield in the CBCT-AF group is very close to that of Pritchett’s and Ali‘s publications, which are nearly 90% [19,27,28]. We believe our results are credible and can be applied in clinical practice. Finally, we do not performed brushing and transbronchial needle aspiration for PPLs diagnosis. Previous publications revealed that TBB combing with these techniques have higher sensitivity [42,43], therefore they were performed routinely in many institution. However, histologic specimen is required for immunohistochemical and molecular tests because of the further cancer management but these techniques only offer cytologic specimen. Due to this reason, we do not employ them for our clinical practice.

## 5. Conclusions

Our study showed that the adoption of CBCT-AF increases the navigation success rate, and thereby facilitates the diagnostic yield of EBUS-TBB for PPLs. CBCT-AF-guided bronchoscopy is associated with an acceptable patient radiation dose and procedure-related complication rate. Due to its safety and accurate navigation, we believe CBCT-AF could be widely applied in bronchoscopic techniques.

## Figures and Tables

**Figure 1 diagnostics-12-00041-f001:**
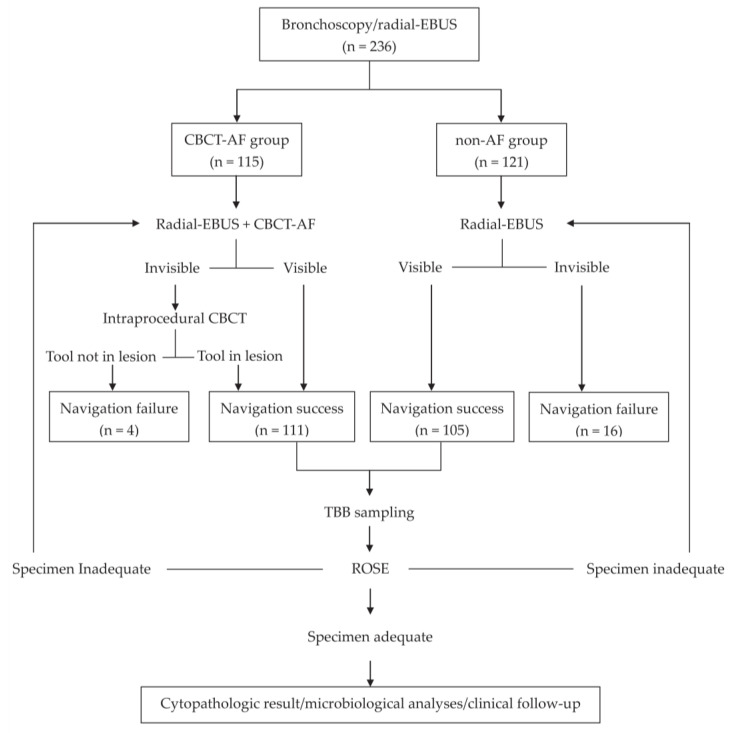
Flowchart of the procedure in CBCT-AF and non-AF groups. AF, augmented fluoroscopy; CBCT-AF, cone-beam computed tomography-derived augmented fluoroscopy; EBUS, endobronchial ultrasound; ROSE, rapid on-site cytologic evaluation; TBB, transbronchial biopsy.

**Figure 2 diagnostics-12-00041-f002:**
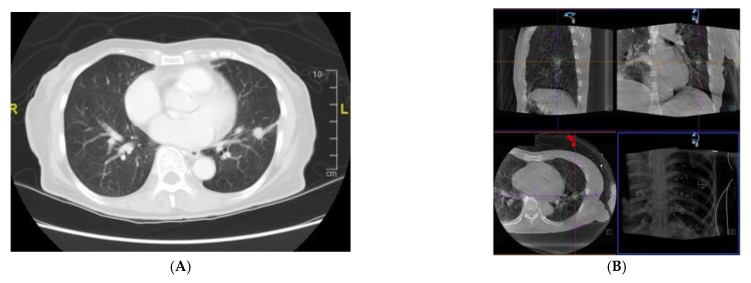
A 75-year-old female patient received CBCT-AF combined with EBUS-TBB, and was finally diagnosed with lung adenocarcinoma. (**A**) standard CT image showing a small nodule (10.6 mm) at the left lower lobe; (**B**) the target lesion on CBCT image contoured in three standard axes for the AF image; (**C**,**D**) a concentric peribronchial lesion discovered by radial-EBUS via AF guidance; (**E**) TBB guided by AF image. AF, augmented fluoroscopy; CBCT-AF, cone-beam computed tomography-derived augmented fluoroscopy; CT, computed tomography; EBUS-TBB, endobronchial ultrasound-guided transbronchial biopsy; TBB, transbronchial biopsy.

**Table 1 diagnostics-12-00041-t001:** Characteristics of the patients.

Characteristics	Total (*n* = 236)	CBCT-AF (*n* = 115)	Non-AF (*n* = 121)	*p-*Value
Age (years old, range)	66.1 (28–94)	65.1 (28–91)	67.0 (31–94)	0.258
Male gender (%)	118 (50)	50 (43.5)	68 (56.2)	0.051
Final diagnosis (%)				
Malignancy	173 (73.3)	92 (80)	81 (66.9)	0.017 *
Lung adenocarcinoma	145 (61.4)	73 (63.5)	72 (59.5)	
Lung squamous cell carcinoma	8 (3.4)	4 (3.5)	4 (3.3)	
Small cell lung cancer	3 (1.3)	2 (1.7)	1 (0.8)	
Other NSCLC	8 (3.4)	5 (4.3)	3 (2.5)	
Metastasis	9 (3.8)	8 (7.0)	1 (0.8)	
Colorectal cancer	2 (0.8)	2 (1.7)	0 (0)	
Breast cancer	2 (0.8)	2 (1.7)	0 (0)	
Urothelial cancer	2 (0.8)	2 (1.7)	0 (0)	
Melanoma	1 (0.4)	1 (0.9)	0 (0)	
Multiple myeloma	1 (0.4)	1 (0.9)	0 (0)	
Hepatocellular carcinoma	1 (0.4)	0 (0)	1 (0.8)	
Non-malignancy	63 (26.7)	23 (20)	40 (33.1)	
Fungus	11 (4.7)	5 (4.3)	6 (5.0)	
Aspergillosis	4 (1.7)	2 (1.7)	2 (1.7)	
Cryptococcus	3 (1.3)	2 (1.7)	1 (0.8)	
Candida	3 (1.3)	0 (0)	3 (2.5)	
Pneumocystis jirovecii pneumonia	1 (0.4)	1 (0.9)	0 (0)	
Mycobacterium tuberculosis	8 (3.4)	2 (1.7)	6 (5.0)	
Non-tuberculous Mycobacteria	2 (0.8)	2 (1.7)	0 (0)	
Pneumonia	15 (6.4)	6 (5.2)	9 (7.4)	
Hamartoma	1 (0.4)	0 (0)	1 (0.8)	
Benign inflammation	25 (10.6)	8 (7.0)	18 (14.9)	

AF = augmented fluoroscopy; CBCT-AF = cone beam computed tomography-derived augmented fluoroscopy; *n* = number; NSCLC = non-small cell lung cancer; * = statistical significance with *p*-value < 0.05.

**Table 2 diagnostics-12-00041-t002:** Cytopathologic results and final diagnosis.

Cytopathologic Finding of TBB	CBCT-AF (*n* = 115)		Non-AF (*n* = 121)	
	*n*	Details of Final Diagnosis	*n*	Details of Final Diagnosis
**TBB diagnostic**	96		90	
Malignant	76		58	
Lung adenocarcinoma	60		51	
Lung squamous cell carcinoma	4		3	
Small cell lung cancer	2		0	
Other NSCLC	4		3	
Metastasis	6		1	
Colorectal cancer	2		0	
Breast cancer	1		0	
Urothelial cancer	2		0	
Melanoma	1		0	
Hepatocellular carcinoma	0		1	
Non-malignancy	20		32	
Fungus	4		3	
Aspergillosis	1		2	
Cryptococcus	2		1	
Pneumocystis jirovecii pneumonia	1		0	
Mycobacterium tuberculosis	2		2	
Non-tuberculous Mycobacteria	1		0	
Granulomatous inflammation	2	Aspergillosis (*n* = 1) Non-tuberculous Mycobacteria (*n* = 1)	1	Mycobacterium tuberculosis (*n =* 1)
Chronic inflammation	11	Pneumonia with tissue culture proved (*n =* 6) Benign inflammation with clinical follow-up (*n =* 5)	26	Candida with tissue culture proved (*n =* 3) Mycobacterium tuberculosis with tissue culture proved (*n =* 2) Pneumonia with tissue culture proved (*n =* 8) Benign inflammation with clinical follow-up (*n =* 13)
**TBB nondiagnostic**	19		31	
No representative samples	5	Lung adenocarcinoma (*n* = 3) Other NSCLC (*n* = 1) Benign inflammation (*n =* 1)	5	Lung adenocarcinoma (*n =* 3) Benign inflammation (*n =* 2)
Chronic inflammation	5	Lung adenocarcinoma (*n =* 5)	9	Lung adenocarcinoma (*n =* 8) Small cell lung cancer(*n =* 1)
Atypical cell	5	Lung adenocarcinoma (*n =* 3) Breast cancer (*n =* 1) Multiple myeloma (*n =* 1)	1	Lung adenocarcinoma (*n =* 1)
Navigation failure	4	Lung adenocarcinoma (*n =* 2) Benign inflammation (*n =* 2)	16	Lung adenocarcinoma (*n =* 9) Lung squamous cell carcinoma (*n =* 1) Hamatoma (*n =* 1) Mycobacterium tuberculosis (*n =* 1) Pneumonia (*n =* 1) Benign inflammation (*n =* 3)

AF = augmented fluoroscopy; CBCT-AF, cone beam computed tomography-derived augmented fluoroscopy; *n*, number; NSCLC, non-small cell lung cancer; TBB, transbronchial biopsy.

**Table 3 diagnostics-12-00041-t003:** Characteristics of EBUS-TBB procedures and target lesions.

Characteristics	Total (*n* = 236)	CBCT-AF (*n* = 115)	Non-AF (*n* = 121)	*p-*Value
Indication (%)				0.100
Initial diagnosis	178 (75.4)	82 (71.3)	96 (79.3)	
Re-biopsy	58 (24.6)	33 (28.7)	25 (20.7)	
Lesion size (mm, range)	27.4 (6.0–64.2)	24.0 (6.0–62.0)	30.6 (6.2–64.2)	<0.001 *
≤20 mm (%)	88 (37.3)	56 (48.7)	32 (26.4)	
Lesion appearance (%)				0.030 *
Solid	193 (81.8)	88 (76.5)	105 (86.8)	
Part solid/GGO	43 (18.2)	27 (23.5)	16 (13.2)	
Location (%)				
Right Upper Lobe	74 (31.4)	48 (41.7)	26 (21.5)	0.001 *
Left Upper Lobe (Left Upper Division + Lingual)	43 (18.2)	16 (13.9)	27 (22.3)	0.066
Right Middle Lobe	25 (10.6)	9 (7.8)	16 (13.2)	0.128
Right Lower Lobe	48 (20.3)	20 (17.4)	28 (23.1)	0.175
Left Lower Lobe	46 (19.5)	22 (19.1)	24 (19.8)	0.511
Presence of bronchus sign (%)	203 (86.0)	95 (82.6)	108 (89.3)	0.099
Location of probe (%)				0.299
Within	177 (75.0)	84 (73.0)	93 (76.9)	
Adjacent to/invisible	59 (25.0)	31 (27.0)	28 (23.1)	
CXR visible (%)				
≤10 mm	2/20 (10)	2/16 (12.5)	0/4 (0)	0.632
>10 mm, ≤20 mm	36/68 (52.9)	23/40 (57.5)	13/28 (46.4)	0.257
>20 mm	132/148 (89.2)	55/59 (93.2)	77/89 (86.5)	0.155
Guide sheath use (%)	124 (52.5)	45 (39.1)	79 (65.3)	<0.001 *
Duration time (min.)	38.1 (12–109)	41.9 (16–109)	34.4 (12–78)	<0.001 *
Intraprocedural CBCT use (%)				-
≤10 mm	-	6/16 (37.5)	-	
>10 mm, ≤20 mm	-	10/40 (25)	-	
>20 mm	-	3/59 (5.1)	-	
Radiation (Gy∙cm^2^, range)				-
Total dose	-	25.48 (9.46–123.41)	-	
Fluoroscopy dose	-	2.70 (0.02–10.31)	-	
Dyna CT dose	-	22.78 (7.26–119.16)	-	
Complication (%)	23 (9.7)	9 (7.8)	14 (11.6)	0.227
Bleeding	8 (3.4)	5 (4.3)	3 (2.5)	0.333
Pneumothorax	3 (1.3)	0 (0)	3 (2.5)	0.133
Fever	12 (5.1)	4 (3.5)	8 (6.6)	0.213

AF = augmented fluoroscopy; CBCT-AF = cone beam computed tomography-derived augmented fluoroscopy; CXR = chest X-ray; GGO = ground glass opacity; *n* = number; * = statistical significance with *p*-value < 0.05.

**Table 4 diagnostics-12-00041-t004:** Comparison of navigation rate and diagnostic yield between the CBCT-AF and non-AF groups.

Lesion Size	Navigation Success Rate		*p-*Value	Diagnostic Yields		*p* Value
	CBCT-AF (*n* = 115)	Non-AF (*n* = 121)		CBCT-AF (*n* = 115)	Non-AF (*n* = 121)	
Total (%)	111/115 (96.5)	105/121 (86.8)	0.006 *	96/115 (83.5)	90/121 (74.4)	0.060
≤10 mm	14/16 (87.5)	0/4 (0)	0.003 *	11/16 (68.8)	0/4 (0)	0.026 *
>10, ≤20 mm	38/40 (95)	20/28 (71.4)	0.009 *	31/40 (77.5)	13/28 (46.4)	0.016 *
>20 mm	59/59 (100)	85/89 (95.5)	0.127	54/59 (91.5)	77/89 (86.5)	0.254

AF = augmented fluoroscopy; CBCT-AF = cone beam computed tomography-derived augmented fluoroscopy; *n* = number; * = statistical significance with *p*-value < 0.05.

**Table 5 diagnostics-12-00041-t005:** Performance of the diagnosis of PPLs between CBCT-AF and non-AF groups.

	Sensitivity	Specificity	PPV	NPV	Accuracy
CBCT-AF					
Total	82.7	100	100	20.8	83.5
≤10 mm	66.7	100	100	16.8	68.8
>10, ≤20 mm	75.7	100	100	25.0	77.5
>20 mm	91.4	100	100	16.7	91.5
Non-AF					
Total	70.2	100	100	35.4	74.4
≤10 mm	0	-	-	-	0
>10, ≤20 mm	37.5	100	100	21.1	46.4
>20 mm	84.2	100	100	52.0	86.5

AF = augmented fluoroscopy; CBCT-AF = cone beam computed tomography-derived augmented fluoroscopy; NPV = negative predictive value; PPV = positive predictive value.

**Table 6 diagnostics-12-00041-t006:** Comparison of diagnostic yields between the CBCT-AF and non-AF groups after excluding the navigation failure population.

Variable	Diagnostic Yield				*p-*Value
	CBCT-AF (*n* = 111)	*p-*Value	Non-AF (*n* = 105)	*p-*Value	
Total (%)	96/111 (86.5)		90/105 (85.7)		0.513
Lesion size (%)					
≤10 mm	11/14 (78.6)		0/0 (0)		-
>10, ≤20 mm	31/38 (81.6)	0.048*	13/20 (65)	<0.001 *	0.141
>20 mm	54/59 (91.5)		77/85 (90.6)		0.547
Indication (%)					
Initial diagnosis	70/78 (89.7)	0.110	72/82 (87.8)	0.202	0.446
Re-biopsy	26/33 (78.8)		18/23 (78.3)		0.607
Lesion appearance on CT (%)					
Solid	73/84 (86.9)	0.521	83/96 (86.5)	0.378	0.554
Part solid/GGO	23/27 (85.2)		7/9 (77.8)		0.475
Position					
Upper lobe	53/61 (86.9)	0.554	36/44 (81.8)	0.245	0.328
Non-upper	43/50 (86)		54/61 (88.5)		0.453
Bronchus sign in CT					
Present	83/95 (87.4)	0.370	88/103 (85.4)	0.734	0.426
Absent	13/16 (81.3)		2/2 (100)		0.686
Position of probe					
Within	74/84 (88.1)	0.282	80/93 (86.0)	0.540	0.428
Adjacent to	22/27 (81.5)		10/12 (83.3)		0.635
CXR visible					
Visible	70/79 (88.6)	0.231	73/85 (85.9)	0.579	0.388
invisible	26/32 (81.3)		17/20 (85)		0.519
Guide sheath					
Use	38/45 (84.4)	0.402	66/76 (86.8)	0.400	0.455
Not use	58/66 (87.9)		24/29 (82.8)		0.355
Final diagnosis (%)					
Malignancy	76/90 (84.4)	0.173	58/71 (81.7)	0.075	0.399
Benign process	20/21 (95.2)		32/34 (94.1)		0.677

AF = augmented fluoroscopy; CBCT-AF = cone beam computed tomography-derived augmented fluoroscopy; CT = computed tomography; CXR *=* chest X-ray; GGO = ground glass opacity; *n* = number; * = statistical significance with *p*-value < 0.05.

**Table 7 diagnostics-12-00041-t007:** The final diagnosis of 14 subcentimeter lesions that were successful navigation.

Diagnosis	*n* (%)
Success TBB diagnosis	11 (78.6)
Malignancy	6 (42.9)
Lung adenocarcinoma	4 (28.6)
Colon cancer	1 (7.1)
Urothelial carcinoma	1 (7.1)
Non-malignancy	5 (35.7)
Mycobacterium tuberculosis	1 (7.1)
Cryptococcus	1 (7.1)
Pneumonia	2 (14.3)
Benign inflammation	1 (7.1)
Failure TBB diagnosis	3 (21.4)
Lung adenocarcinoma	2 (14.3)
Benign inflammation	1 (7.1)

*n* = number; TBB = transbronchial biopsy.

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
