# Peer review of "Cone-Beam Computed Tomography-Derived Augmented Fluoroscopy Improves the Diagnostic Yield of Endobronchial Ultrasound-Guided Transbronchial Biopsy for Peripheral Pulmonary Lesions"

_diagnostics, 2021, doi:10.3390/diagnostics12010041_

Round 1

Reviewer 1 Report

1. The manuscript is too difficult to read. The reason is that the method of comparing/explaining the newly developed tool with the existing one is usually very different from usual.

2. Case-control randomization was not done. This would be a limitation of retrospective study, but some of the key factors that can affect the results should be macthed through the well known methods such as Propsensity score matching (PSM). This should minimize postive findings that are actually meaningful but could be missed.

3. In the materials and methods, a diagram showing the overall procedure process is needed. It is necessary to clearly indicate at what point the difference between Case and Control exists.

4. In the materials and methods, 2.3 Diagnostics, explanation on "Gold standard" or "Reference sample" is requiried.

5. The metric of the each test was not appropriate. Tables should be prepared with commonly used metrics such as sensitivity, specificity, and accuracy.

6. When comparining diagnotic yields of 2 methods, use compare AUC methods.

Reviewer 2 Report

This article is of interest to readers. However, it needs some modifications in its current state. 

  1. The information on rapid on-site evaluation during the procedure should be detailed (e.g.,Did you use Diff-Quik stain during ROSE?)
  2.  In Table 3, there was no information about the visibility of lesions on chest x-ray between the two groups; CBCT-AF may contribute to improve the diagnostic accuracy of lesions that were not detected on chest x-ray.
  3. There was a significant difference in the rate of guide sheath use between the two groups. Why did such a difference occur? When comparing the diagnostic accuracy between the two groups, the sampling method without the use of guide sheath should be mentioned in the text.
  4. I don't think there was a title for Table 5. An appropriate title should be provided for Table 5.
  5. In the small lesions (<10mm) in Table 5, the diagnostic yield of the CBCT-AF group was significantly higher than that of the Non-AF group. What were the histological findings of these small lesions? In previous reports, it was very difficult to diagnose small lesions (less than 10 mm) via R-EBUS and navigation system. The information on what kind of histological diagnosis was obtained by Cone-Beam Computed Tomography-Derived Augmented Fluoroscopy in this study may be useful for clinicians involved in the diagnosis of PPL.

Round 2

Reviewer 2 Report

The revised paper was well written. Therefore, this paper seems to be worth publishing.